# Recent Advances in Origins of Life Research by Biophysicists in Japan

**Tony Z. Jia** [1,*]  **and Yutetsu Kuruma** [1,2]

[1]  Earth-Life Science Institute, Tokyo Institute of Technology, 2-12-1-IE-1 Ookayama, Meguro-ku, Tokyo 152-8550, Japan; kuruma@elsi.jp

[2]  JST, PRESTO, 4-1-8 Honcho, Kawaguchi, Saitama 332-0012, Japan

*   Correspondence: tzjia@elsi.jp; Tel.: +81-03-5734-2708

**Abstract:** Biophysics research tends to focus on utilizing multidisciplinary technologies and interdisciplinary collaborations to study biological phenomena through the lens of chemistry and physics. Although most current biophysics work is focused on studying extant biology, the fact remains that modern biological systems at some point were descended from a universal common ancestor. At the core of modern biology is the important question of how the earliest life on (or off) Earth emerged. Recent technological and methodological advances developed by biophysicists in Japan have allowed researchers to gain a new suite of knowledge related to the origins of life (OoL). Using these reports as inspiration, here, we highlight some of the significant OoL advances contributed by members of the biophysical research field in Japan with respect to the synthesis and assembly of biological (or pre-biological) components on early Earth, the co-assembly of primitive compartments with biopolymer systems, and the evolution of early genetic systems. We hope to provide inspiration to other biophysicists to not only use the always-advancing suite of available multidisciplinary technologies to continue their own line of work, but to also consider how their work or techniques can contribute to the ever-growing field of OoL research.

**Keywords:** biophysics; origins of life; astrobiology; interdisciplinary science

## 1. Introduction

The origins and evolution of life (both on Earth and off of Earth) is one of the biggest mysteries in modern science. In order to understand this fully, we must answer a multitude of questions from a variety of fields, including how the first elements were produced, how the universe emerged, how planets formed, how the Earth's atmosphere and geology developed, how the first organic molecules appeared on Earth, how these molecules assembled and polymerized into the first genetic biopolymers, how these biopolymers evolved into functional biopolymers encapsulated by a membrane compartment to form the first cells on Earth, and how these primitive cells evolved into the last universal common ancestor. Each of these topics provide a small piece of the puzzle that is the origins of life (OoL), and a multitude of impressive and incredible discoveries have continued to take place in each of these areas throughout the last few decades [1–3].

During the outset of OoL research starting in the 1950s, researchers mostly focused on chemical approaches to find out what organic molecules could be created in prebiotic environments, such as in meteorites [4,5] or from atmospheric discharge experiments [6–8], through chemical analyses of simulated prebiotic environments or real samples. These analyses were revolutionary and provided much of the basis of what organic chemicals we believe today to have been present on early Earth, i.e., which chemicals would have been available to participate in reactions that could ultimately lead to the assembly of the first genetic and/or functional biopolymers on early Earth. In both of these cases,

the required chemical analyses and setups were fairly simple, and in fact the original samples in both the Murchison meteorite (a carbonaceous chondrite (a stony, non-metallic meteorite) that fell in rural Western Australia in 1969, which has been used as the basis for determination of the prebiotic chemical inventory on early Earth) [9] and a 1958 Miller spark discharge experiment [10,11] were reanalyzed more recently with modern technological and methodological advances. These reanalyses resulted in the detection of many more organic molecules than originally detected, leading to a reimagining of the chemical environment on early Earth.

In the 60 plus years that have followed the initial study by Miller, we have also learned about how first genetic biopolymer on Earth, RNA, could have replicated non-enzymatically [12]; how primitive cells can assemble from simple fatty acids [13,14]; and how other simple molecules, such as amino acids or polyesters, could have easily been synthesized through seasonally-effected wet-dry cycles [15,16]. In addition, large radio telescopes can now inventory the molecules in certain space systems [17], geochemical techniques can be used to better simulate prebiotic reaction conditions [18], artificial cells that can replicate genetic material can now be created in the laboratory using a mixture of chemistry, biology, and physics techniques [13,19], and materials science techniques can be used for the study of self-assembling systems relevant to OoL [20,21]. Each of these discoveries was facilitated by interdisciplinary collaborations and advances in technology and methodology, suggesting that the future of OoL research will continue to rely heavily on new technologies and interdisciplinary cooperation.

One such field, which by definition is interdisciplinary, and also places an emphasis on studying traditional biological systems through new, never before used techniques is biophysics. Biophysics generally uses physical and chemical tools to study biological systems. These multidisciplinary approaches have led to many new discoveries in biology that traditional biology have not been able to do without the aid of multidisciplinary tools and interdisciplinary collaborations. These include (but are certainly not limited to): Using super resolution imaging to characterize the structure of brain synapses [22], using cryo-electron microscopy (cryo-EM) to solve structures of non-crystalline compounds [23], using advanced DNA sequencing techniques to quantify low-copy noncoding RNAs [24], and incorporating nonequilibrium stochastic calculations to study gene expression fluctuations [25]. Each of these techniques was used to uncover a mystery regarding a modern biological system, of which many mysteries still remain and many more may appear as more knowledge is gleaned. As such, continuing to use biophysical tools and novel interdisciplinary techniques to study extant biology is certainly a worthy endeavor.

One of the remaining questions at the core of biology is how these modern systems arose in the first place. As early terrestrial life forms could have been based on principles inherent to extant life, studying OoL through the focus of biophysics is a reasonable way to approach this field of research. Recently, there has been a variety of OoL-relevant research within the biophysics research community in Japan: from imaging to combinatorial biology to computational biology to biochemistry to artificial cell synthesis to synthetic biology, etc. Researchers studying pre "life" can still use biophysical tools, as the techniques that are used in modern biophysical studies are broadly applicable to many living and non-living systems. Although it is possible that the original intent of the researchers was not to specifically study OoL, nevertheless, many of these studies are at the forefront of OoL research. Here, we highlight the contributions of certain members of the biophysics research field in Japan towards advancing OoL research, specifically towards the synthesis and assembly of the first biological components on early Earth, co-assembly of primitive compartments with biopolymers, and evolution of early genetic or catalytic systems (Figure 1). We then conclude with a prospective on the future of OoL research, and how biophysicists can incorporate novel biophysics technologies with continued international and interdisciplinary collaborations to push OoL research forward.

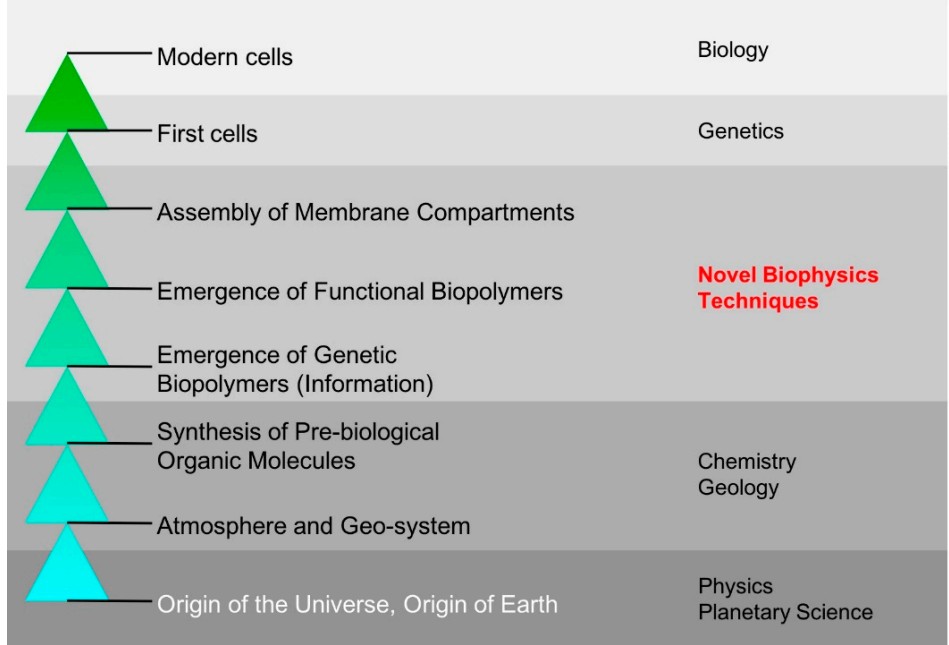

**Figure 1.** A chronological view of origins of life research (from bottom to top), starting from the origin of Earth and ending at modern cells. Specifically, biophysicists in Japan have been contributing towards knowledge on the emergence of the first genetic and functional biopolymers as well as the assembly of the first membrane compartments on early Earth; these are believed to be important steps in the origins of life. As the biophysics research community in Japan has a good track record in developing new techniques and methodologies for biological analysis, by incorporating these new techniques along with a renewed focus on more international and interdisciplinary collaborations, the entire origins of life community can glean even more knowledge in these poorly-understood, but incredibly important, facets of the origins of life. Traditionally, the most advances in each problem (left) have been supplied by researchers in the listed fields on the right. However, given the gradual advancement of research into an interdisciplinary effort, we will not limit the solving of these major problems to one specific field. Additionally, understanding the origins of life will inherently require an understanding of all of the problems listed, and then some, and thus only through collaborative efforts of all fields listed, including biophysics, can the origins of life be fully (or at least better) understood.

## 2. Synthesis, Assembly, and Regulation of Primitive Biopolymers

The modern central dogma consists of a multitude of moving parts moving in concert to convert DNA to RNA to proteins, each step of which is catalyzed by certain enzymatic components. This process exists in each living organism (we will leave a discussion on viruses for another time [26,27]) and recent discoveries in Japan have shown that it is possible to engineer the components of the central dogma to either express non-peptide components, such as polyesters [28], to reduce the genetic code to less than 20 amino acids [29–32], or to introduce the ability to add non-natural amino acids into translated peptides [33]. Each of these pioneering studies provided a way for the synthetic biology research community to reimagine the way that modern biological machinery could be utilized for clinical and biotechnological applications. However, if the central dogma protein expression pathway can still be run with non-peptide products or with a smaller or larger suite of available amino acids than the 20 seen in modern biology, why exactly were specifically DNA [34], RNA, and peptides (amino acids) chosen in this order by evolution over all of the other possible biomolecules in order to form the modern genetic code [35]? Biological polymers in modern life also are completely composed of a specific handedness (ʟ-peptides and ʀ-nucleic acids) [36], however, recent reports have also shown that a replicating system could be viable even with the existence of the mirror-image biopolymers [37] or even mixed chirality systems [38]. Thus, it is inherently important to understand how and where nucleic acids and amino

acids/peptides could have arisen and assembled on the early Earth and why these molecules (and their specific homochiralities) were chosen by evolution. Recent prebiotic chemistry advances suggest the existence of plausible pathways on the early Earth by which amino acids and nucleotides could have been synthesized on the early Earth [39,40]. However, specific reactants, catalysts, and environments are required for such reactions to occur.

In almost all cases, some type of catalyst or energy source is required to generate new high-energy covalent bonds. *Ab initio* calculations performed by the group of Shigenori Tanaka of Kobe University have suggested that meteoritic impacts could have provided the energy required to generate larger moieties from simple, commonly-available organic precursors, such as formic acid (and other carboxylates and Fischer-Tropsch synthesis products), from carbon dioxide and water [41], or ammonia from nitrogen and water [42]. Formic acid and ammonia are themselves significant molecules on early Earth as combining both of these molecules is one way to produce formamide (although there are no known sources of formamide on Earth today, it has also been detected in extraterrestrial sources as well and can be produced in a variety of other methods other than that discussed here [43]), itself an incredibly important prebiotic molecule that forms the basis of much of the prebiotic chemistry leading towards synthesis of nucleotides and amino acids [44,45].

Other important prebiotic catalysts include mineral surfaces, which provide not only increased local concentrations and electrochemical driving forces, but can also result in favorable orientational specificity of adsorbed molecules [46]. To date, much work has focused on harnessing these catalytic properties to study nucleic acid [47,48] and peptide polymerization in non-enzymatic mineral-catalyzed settings, but these mineral surfaces could also provide a protective or stabilizing property for these biopolymers (or components thereof) [49]. Yoshihiro Furukawa of Tohoku University (and co-workers) discovered that borate minerals selectively provide chemical stability to ribose sugars, which are normally one of the most unstable pentose sugars, by forming a ribose–borate complex that inhibits ribose hydrolysis (Figure 2). This suggests that borate mineral binding may be one method by which ribose was initially selected as the sugar moiety in nucleotides as opposed to other pentoses [50]. Additionally, combining ribose-binding minerals with formamide-based prebiotic chemistries and nucleosides results in phosphorylated nucleotide products with the correct regiochemistry as compared to modern nucleotides [51], providing another plausible pathway by which primitive biomolecules could have been synthesized on Early earth.

**Figure 2.** Formation of ribose–borate complexes which stabilize ribose by inhibiting the hydrolysis of ribose (by environmental factors). Reprinted with permission from Furukawa, Yoshihiro, Mana Horiuchi, and Takeshi Kakegawa. 2013. "Selective Stabilization of Ribose by Borate." Origins of Life and Evolution of the Biosphere: The Journal of the International Society for the Study of the Origin of Life 43 (4–5): 353–361 [50]. Copyright 2013 Springer Nature.

Not only do mineral surfaces catalyze prebiotic chemical reactions and polymerization of primitive biopolymers, they also effect the self-assembly of molecules to form functional structures with complex emergent properties without the need for formation of new high-energy covalent bonds (instead utilizing spontaneous thermodynamically-favorable interactions, such as van der Waals interactions, hydrogen bonding, and hydrophobic interactions, to name a few). These catalyzed phenomena include changes in nucleic acid secondary structure, assembly of organic monolayers, and formation of

peptide amyloids [46]. In particular, peptide amyloid fibrils, which are assembled from a hydrophobic peptide found in Alzheimer's disease patients [52], although themselves not prebiotically plausible due to their length (42 amino acids), provide a model system by which other potential prebiotic self-assembling peptide fiber systems (which are much shorter in length), such as tripeptides, like KYF (K: Lysine, Y: Tyrosine, F: Phenylalanine) [20], can be studied. Recent advances in high speed atomic force microscopy (AFM) made by the Nano Life Science Institute at Kanazawa University allow high time-scale resolution studies of the dynamics of amyloid peptide fibers [21] (and other prebiotic self-assembling peptide fibers) [53], while mechanisms of amyloid peptide fiber hydrolysis by short peptides have been discovered by researchers at Kobe University [54,55]. Although these amyloid peptide fiber hydrolysis studies are mainly applicable to clinical treatment research of Alzheimer's patients, understanding how short peptides can catalyze the assembly, transition, and/or disassembly of any biochemical or biopolymer structure would shed light on how short peptides could have contributed to the regulation of prebiotic chemical reactions that eventually led to the first biochemical regulatory interactions on early Earth.

In fact, relevant systems studied by biophysicists would also include not only studies on small peptides, but also other prebiotically-available structures, like polyamines, which can induce structural changes in DNA [56], or peptide-like systems, such as proteins (which again are not prebiotically plausible themselves, but as many protein-effected processes are mainly catalyzed by a few key residues, it is not out of the realm of likelihood that similar processes could have been accomplished by one or many short peptides), which have been shown to drive changes in self-assembled RNA nanostructures for nanotechnology purposes or for studying cellular regulation processes [57–60]. Some RNA self-assemblies have themselves been shown to be catalytically active (in fact, they may be more efficient than a monomer ribozyme component) [61–63] and the discovery of a peptide-based (or any other prebiotically available molecule) regulatory mechanism of primitive ribozyme self-assemblies could have a major impact on how we view primitive chemical and genetic evolution. Just like this, the lessons and principles gleaned from these and other non-prebiotic systems can and should be applied towards furthering our understanding of the emergence of the first biomolecules.

## 3. Assembly and Co-Assembly of Primitive Compartments

As modern cells are composed of lipid bilayer membranes which encapsulate genetic and metabolic materials and allow inflow and outflow of nutrients and waste through membrane channels [64], so too has it been long understood that compartments are essential for the emergence of the first cells [65]. A compartment boundary would have been especially important for emerging primitive cellular systems as it would increase the concentration of certain reactants to promote important chemical reactions [66], prevent encapsulated genetic components from diffusing away and allowing for selective evolution of genetic replicators within the encapsulated space [67], and preventing the proliferation of parasites [68,69]. To that end, fatty acid bilayer membrane vesicles have been long postulated to have been the first protocellular compartments due to their spontaneous assembly from micelles or films at physiological conditions, as well as their impermeability to longer genetic molecules (i.e., RNA) while at the same time being permeable to smaller nutrients [70]. Furthermore, phospholipid vesicles have also been used as model systems as they exist in a temporal space in between primitive fatty acid membrane vesicles and modern phospholipid membrane cells [71]. Coupled with the recent technological explosion in Japan for technologies that assist in assembly and modification of vesicles and lipid membranes [72–77], many researchers in Japan have been able to contribute greatly towards relevant research on studying the formation, dynamics, and potential transitions of phospholipid vesicles, study of membrane lipid flip-flop dynamics in asymmetric vesicles by researchers at the Institute of Industrial Science at University of Tokyo [72], induction of vesicle division by the introduction of non-vesicle-forming lipids by researchers at Nara Institute of Science and Technology [78], and complex geometries of pore-forming, adhesive, and dividing vesicles by researchers at Tohoku University [79], just to name a few.

While understanding vesicle assembly and dynamics has been a major step forward in the field of artificial cell synthesis and understanding protocellular origins, we still lack the understanding of how primitive cells could have induced their own growth, replication, and proliferation in a controlled manner. One of the major facets of life is the modern cell cycle, which is the mechanism by which cells control their growth, replication, and proliferation (by division) [64]. In the absence of such control, cells could fail to grow, or even result in uncontrolled growth, such as that which is seen in cancer cells [80]. While the most primitive of cells likely could have taken advantage of environmental processes, such as changes in salinity or pH due to seasonal or diurnal wet-dry cycles [81,82] for assembly and disassembly [83], or utilizing turbulent water flows from underwater vents [84] for division due to shear stress [85], it would still have been essential for primitive cells to have evolved or developed more tractable regulatory mechanisms independent of random or systematic meteorological fluctuations for essential cellular processes. Short of synthetically engineering an artificial cell system with genetic and metabolic machinery capable of producing a cell-cyclical behavior [86,87], communication [88], or autotrophy and energy capture [89,90], as has been reported by members of the artificial cell synthesis field in Japan, there is now a significant push for research to focus more on interactions between prebiotically available (or controllable) biomolecules and biopolymers and lipid bilayer vesicles, or their associated co-assemblies.

Recent work by researchers in Japan have focused on interactions between lipid vesicles and genetic materials, i.e., DNA. Researchers at Kyoto University described a system wherein a mixed DNA/phospholipid film could spontaneous form giant unilamellar vesicle assemblies with encapsulated DNA, controlled by rehydration [91]. Shimobayashi and Ichikawa dried DOPC vesicles and DNA on a glass slide into a film, and observed tube-like structures, eventually leading to segregated DNA and lipid structures, which depended on the size and concentration of the DNA component (Figure 3A–C). Rehydration of the mixed film led to rapid formation of unilamellar vesicles, while in the absence of the DNA, it resulted only in multilamellar vesicles (Figure 3D–F). It was postulated that osmotic pressure differences generated by the segregated DNA phase resulted in additional repulsive forces for the lipid layers and the formation of unilamellar, rather than multilamellar, vesicles. This work shows not only that vesicles can efficiently encapsulate genetic materials after rehydration from a film-like state, but also suggests a positive-feedback symbiotic relationship by which the genetic materials actually promote their own encapsulation in giant unilamellar vesicles, which could lead towards their own increased ability to evolve within a vesicle compartment.

Another important DNA-lipid vesicle interaction that was recently reported was the development of a DNA nanostructured cytoskeleton for phospholipid vesicles by a large conglomerate of researchers from many Japanese institutes [92]. It is well known that lipid vesicles are far less physically and mechanically stable to various environmental shocks, such as osmotic shock, than modern cells. This is due to the existence of an actin cytoskeleton beneath the modern membrane bilayer. Thus, by engineering DNA in such a way to self-assemble into specific network architectures, the team of researchers was not only able to form spherical DNA compartments in bulk, but also incorporate these DNA self-assemblies as cytoskeletal components underneath a phospholipid (mixed 1,2-dioleoyl-3-trimethylammonium-propane (DOTAP), 1-palmitoyl-2-oleoyl-sn-glycero-3-phosphocholine (POPC), and mineral oil) vesicle. This resulted in a hybrid vesicle system with more stable structural properties than before. Although both of these above-mentioned studies used DNA, they are still very important examples of symbiotic and regulatory interactions between primitive vesicle compartments and primitive genetic materials; one can of course further extend this work into the realm of the more prebiotically-relevant RNA.

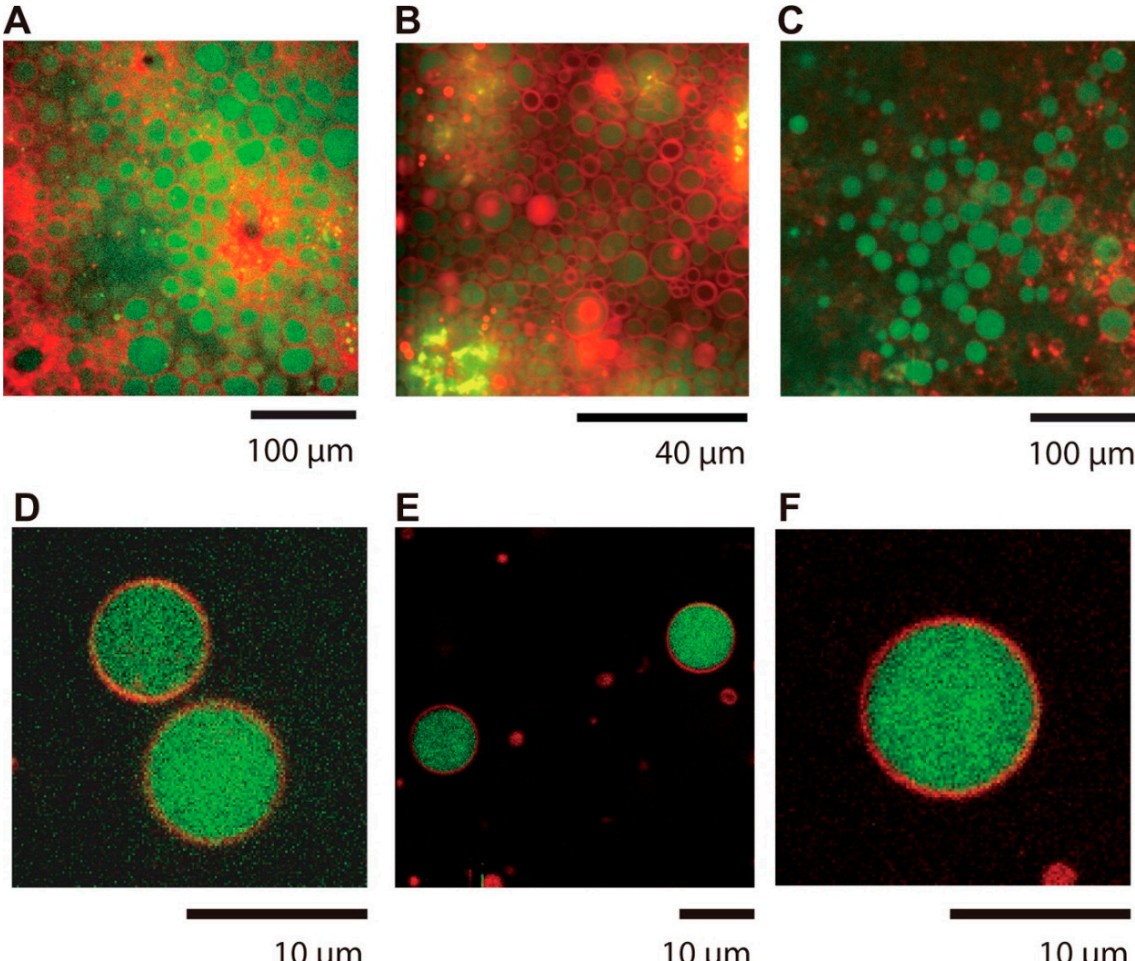

**Figure 3.** Phospholipid (red; Rho-PE stain) vesicles encapsulating DNA (green; YOYO-1 stain) on a surface (**A**–**C**) and after rehydration into solution (**D**–**F**). Scale bars as described (**A**,**D**) fragmented salmon DNA, (**B**,**E**) λ phage DNA, and (**C**,**F**) T4 phage DNA. Reprinted with permission from Shimobayashi, Shunsuke F., and Masatoshi Ichikawa. 2014. "Emergence of DNA-Encapsulating Liposomes from a DNA-Lipid Blend Film." The Journal of Physical Chemistry. B 118 (36): 10688–10694 [91]. Copyright 2014 American Chemical Society.

In addition to genetic components, catalytic components, such as short peptides, also have significant regulatory effects on lipid vesicles. Small peptides as short as two residues in length have been shown to affect the growth of vesicles [19], while arginine- and lysine-containing peptides have been shown to act as cell-penetrators (which have applications in drug delivery and discovery applications [93,94]), and completely synthetic transmembrane barrel peptides have even been incorporated into lipid membranes [95]. The lab of Masahito Yamazaki in Shizuoka University has been researching cell penetrating peptides into vesicles and actual cells [96]. In certain cases, including when studying the 21-mer antimicrobial cationic peptide, PGLa [97], or the 21-mer cell penetrating peptide, TP10 [98], they discovered that in both cases, the peptide induces pore formation in mixed dioleoylphosphatidylglycerol (DOPG) and dioleoylphosphatidylcholine (DOPC) phospholipid vesicle membranes, causing leakage of the internal components, which when applied to real bacterial cells, results in cell death. The result is translocation of the peptide into the interior of the vesicle. However, in other cases, such as in the 6-mer LfcinB (4–9) peptide, translocation into the vesicle interior did not result in pore formation or encapsulated component leakage [99]. As short peptides are very abundantly available on early Earth, these results suggest the large potential for further studies of primitive short vesicle-penetrating peptides as a way to control protocellular dynamics, for example,

the exchange of membrane lipid components [100], protocellular growth, or even cell death. Similar to the symbiotic relationship with DNA, where DNA nanostructures can be assembled on lipid vesicle surfaces [101,102], vesicle–peptide interactions could also have been an important component in primitive protocellular development, as recent reports have also shown that lipid vesicles can promote amyloid peptide assembly [103]; an extension of these studies into more prebiotic avenues may result in the discovery of novel primitive structures with emergent properties due to the symbiotic cooperation of the components.

However, in spite of all of this focus on lipid vesicles, primitive compartments need not necessarily be lipid bilayer vesicles, as there are many other model systems that could have afforded essentially the same properties as lipid vesicles, including amphiphilic peptide vesicles [104], cationic organic amphiphile membrane vesicles [87], inorganic chemical cells [105], oil-in-water microdroplets [69], coacervate [106–108] (Figure 4) and aqueous two-phase system [109,110] droplets, and even membraneless polyester microdroplets [111], each of which is important for not only biophysicists, but also materials scientists. We look forward to the further collaboration between these two fields in Japan and abroad not only for further elucidation of the assembly and dynamics of primitive compartments in the context of OoL research, but also perhaps for the development of new therapeutic tools or new discoveries about modern biological systems as well [46].

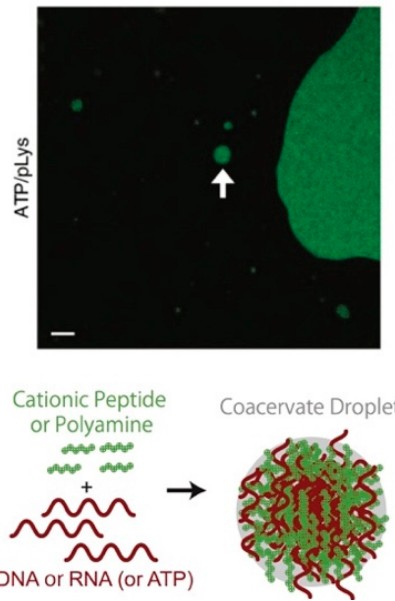

**Figure 4.** Example of a coacervate droplet system composed of a nucleotide (ATP) and a cationic peptide (poly-lysine; pLys) (10 μm scale bar), resulting in visible liquid-liquid phase-separated compartments. Top image reprinted with permission from Jia, Tony Z., Hentrich, Christian, Szostak, Jack W. 2014. "Rapid RNA Exchange in Aqueous Two-Phase System and Coacervate Droplets." Origins of Life & Evolution of the Biospheres 44 (1): 1–12 [108] under a Creative Commons License.

## 4. Primitive Biopolymer Evolution

One of the major components required for an emerging biological system is a self-sustainable evolving genetic system, which eventually would in turn lead to an expression or display of this "genotype" into a functional "phenotype" [112]. Much work within the field of OoL has focused on the selection of RNA ribozymes and aptamers through *in vitro* evolution experiments [113], which simulate external driving forces and selective pressures that result in the "survival of the fittest" ribozyme or aptamer. Reviewing all of the major engineered proteins, peptides, and ribozymes discovered through *in vitro* evolution (many of which have been contributed from the biophysics research community in Japan [29,114–117]) is beyond the scope of this review (and frankly, would be too much to cover in

a single targeted review anyways). Instead, we will focus mostly on novel techniques or methodological advances resulting in studies that contribute further to our knowledge of the initial emergence of evolvable chemical systems.

Genetic evolution generally selects for the sequences which are the most fit with respect to a selective pressure, and thus as researchers, we often view this as a way to select for the sequence with the highest activity or binding (in the case of an aptamer). However, there are also other parameters by which sequences could be selected even through purely environmental methods. For example, in thermophoretic pools, such as those that could occur in small rock pores in oceans or lakes on early Earth [118], Maeda and co-workers found that RNA with secondary structures containing longer stems were more likely to accumulate than those with shorter stems [119]. Of course, this type of selection through a physical property also complements existing knowledge that the emergence of double-stranded RNA stem-loops would have been important for early evolving systems as they contribute to a variety of important biological properties, such as active site recognition, folding, or protection from degradation, just to name a few [120]. Similarly, it has also been shown by Mizuuchi and co-workers that mineral surfaces, ubiquitous on early Earth, are able to selectively bind and select longer RNA strands as opposed to shorter strands, especially at higher temperatures [121]. As longer strands likely have more potential to fold into catalytic or functional structures (due to the larger sequence space), a mineral-based selection could give a further "push" to an early evolving system to select for these longer structures. In addition, in each of these cases, as well as other primitive evolutionary systems, the initial library diversity must be taken into account. The conundrum is akin to Goldilocks, who had to have porridge that was "just right", rather than too hot or too cold; applied to evolutionary purposes, Mizuuchi and co-workers showed that initial libraries with too low or too high sequence diversity were not amenable to efficient replication (and subsequently, evolution) [122]. Instead, it is intermediate sequence diversities that promote RNA networks capable of reproduction (self- or cross-) and the formation of collectively autocatalytic sets. Thus, it is very reasonable to believe that some of the larger evolutionary leaps in early genetic systems could have occurred in such environments if the "physical property" selective pressures (i.e., thermophoresis or mineral binding) contribute to limiting the sequence diversity to that of an intermediate size.

However, in the cases of evolutionary selection of genetic components in various environmentally-promoted conditions, one possible problem that an early evolving genetic system would have had is the emergence of parasites, or "cheaters" [123]. These are sequences whose self-replication is so efficient that their prolonged efficient replication overwhelms the system and chokes out all other sequences, preventing their replication (and subsequent evolution) [124]. Such parasitism would effectively poison a system, and thus discovering methods by which primitive evolving systems could overcome parasitism has been a key focus of the field. Norikazu Ichihashi's lab (now at University of Tokyo) discovered that compartmentalization (as described above) is one way by which parasitic genetic materials (in their case, RNA) can be neutralized [68]. In their seminal study, the researchers designed a system where translation of a host RNA coded for the production of an RNA replicase enzyme (Figure 5). This enzyme would be able to catalyze further production of the initial RNA. However, there also existed a parasitic RNA (which also replicates by the same replicase enzyme) in the system with an identical sequence as the initial host RNA, just without the replicase coding region. They discovered that in bulk, the parasitic RNA completely overwhelmed the host RNA, leading to proliferation of only the parasitic RNA. However, when the system was compartmentalized in an oil-in-water microdroplet, an oscillatory replication pattern emerged, with the parasitic RNA never overwhelming the host RNA. In fact, over time, the host RNA even acquired some level of immunity from the parasitic RNA. The compartments here apparently shelter the host RNA from the parasitic RNAs, as rapidly replicating parasitic RNA within one compartment quickly saturate that compartment. Yet, these parasites are still confined to only their original compartment; parasites can never be transferred into any other compartment (at least in this system, which is a reasonable analog for replicating RNA in lipid vesicles as well), which allows the host RNA to continue to replicate at its

own pace unhindered by the presence of any parasites. Furthermore, co-replicating and co-evolving sets of RNA also are able to develop some resistance to intermediate concentration parasites once a catalytic replicating network has emerged within the system [125].

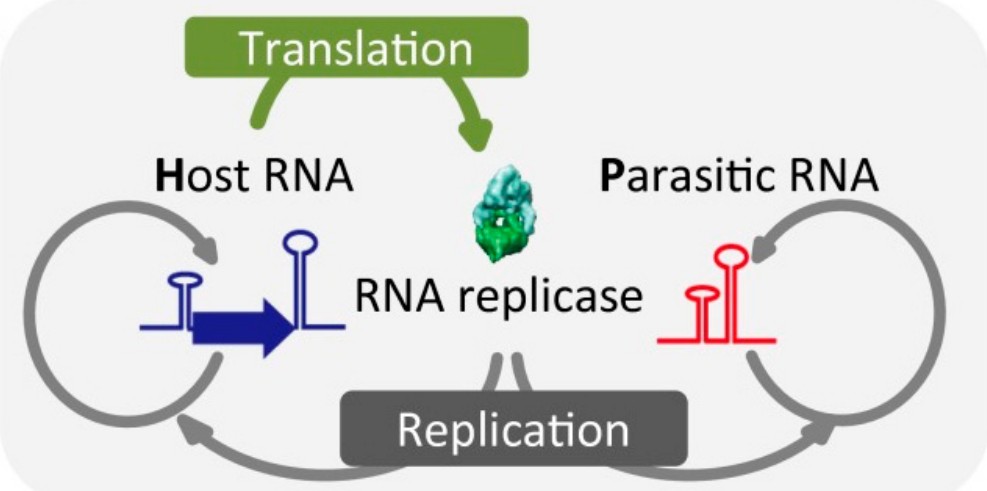

**Figure 5.** Replication scheme described in [68]. Briefly, a host RNA (blue) contains a sequence which codes for translation of an RNA replicase enzyme. This RNA replicase in turn catalyzes the replication of the host RNA. However, a parasitic RNA (which also replicates by the replicase enzyme), but without the replicase coding region, also exists in the system. Reprinted with permission from Bansho, Yohsuke, Taro Furubayashi, Norikazu Ichihashi, and Tetsuya Yomo. 2016. "Host-Parasite Oscillation Dynamics and Evolution in a Compartmentalized RNA Replication System." Proceedings of the National Academy of Sciences of the United States of America 113 (15): 4045–4050 [68]. Copyright 2016 National Academy of Sciences.

While the previous examples highlighted methodological advances which greatly contributed to our knowledge on early genetic evolution, the evolution of peptides and proteins is still of interest both pre- and post-Central Dogma emergence. Although many *in vitro* evolution techniques, such as mRNA display [126] or phage display [127,128], have been developed to study peptide and protein evolution, these are mostly limited to soluble non-membrane proteins. Membrane proteins are incredibly difficult to express due to their specific environmental and expression requirements [128], leaving a void in our understanding of the function and evolution of many membrane proteins. In order to ameliorate this gap in knowledge Yomo and co-workers recently developed a novel technology which allows for *in vitro* screening and evolution of membrane proteins called liposome display [129]. In this technique, a single DNA molecule from a random library is encapsulated within a single POPC (or other phospholipid) vesicle and is transcribed and translated into proteins through an *in vitro* translation system, resulting in the expression of a single protein in each single vesicle. As these proteins have been synthesized within a membrane-containing structure, they can then be incorporated readily into the membrane itself, resulting in membrane proteins being stabilized within the membrane. Then, subsequent functional assays and selections can be performed; in the case of Fujii et al., an alpha hemolysin pore-forming protein was selected for through fluorescence detection and fluorescence-activated cell sorting (FACS), resulting in an evolved protein with a pore-forming activity greater than 30-fold higher than the wild type protein, while at the same time generating important information regarding the evolutionary trajectory of a membrane protein previously not well-characterized [130]. Other non-membrane proteins of interest, such as an aminoacyl-tRNA synthase [131], have also been evolved in this way, paving the way for wider use among the study of protein evolution at-large. Continued development of other novel, creative peptide, protein, or RNA evolution techniques by the biophysics research

community will continue to allow greater accessibility to understanding the emergence and evolution of classes of biopolymers that have not been well-studied.

While experimental evolution and replication studies allow researchers to study evolutionary processes in real systems (both in prebiotic-model systems as well as systems which make no effort to model a primitive system), there are still experimental limitations that can be probed. For example, it is still not possible to sample the entire sequence space of a random nucleotide 30-mer, or even a random peptide 15-mer; there are simply too many possible sequences. Additionally, real-time experimental evolution and replication experiments take time, which is something that is not infinite for researchers. One major question in the experimental OoL field is the fact that the systems used and the reactions we perform in the laboratory are likely significantly shorter timescale-wise than actual processes on prebiotic Earth and thus may not be appropriate systems for probing real geological and prebiotic chemical processes. However, theoretical and computational research is able to probe long evolutionary or replicatory processes (which cannot be reasonably measured experimentally) and give insights into general principles of evolution and replication, some of which have informed experimental researchers on the most optimal or best design strategy for experimental evolutionary systems.

Theory and computation are thus two very important aspects of origins of life research, and two of the main pioneers in theoretical evolution studies are Kunihiko Kaneko and Takashi Ikegami of the University of Tokyo [3,132,133]. Recent theoretical work by the Kaneko Laboratory has featured optimizing parameters for self-replicating polymer systems [134], designing replicatory protocells [135], and identifying the effect on selection pressure magnitude on evolutionary systems [136], just to name a few. In fact, many of their initial studies on origins and evolution of life are relevant to artificial life systems, both in silico and in real systems, resulting in a large amount of their research specifically on the topic of what characteristics define life and studying the emergence of these complex life-defining phenomena, such as signaling [137] or motility [138]. The Ikegami Laboratory has continued to expand this work and have begun to study complex behaviors observed in real systems that are likely the result of billions of years of evolution. These include complex phenomena, such as the emergence and organization of collective motion (flocking or swarming in birds, or fish, or insects) [139,140], the emergence of consciousness or perception [141,142], and the control or coordination of neuronal networks [143,144]. In fact, many of these studies focused not just on the properties or the parameters leading to the emergence of these phenomena, but also probed the evolution of these complex phenomena. Thus, it is imperative to sustain continued collaborations between theoretical and experimental researchers studying biopolymer or genetic evolution in biophysics as well as in other areas within the scope of OoL.

## 5. The Future of OoL Research: Incorporating Novel Biophysics Technologies and Methodologies

The biophysics research community in Japan has contributed many pioneering studies towards furthering our understanding of the origins of life, from how meteoritic impacts on early Earth could have supplied energy and molecules leading to early biochemistries [41,42], all the way to understanding the evolution and activity of bacteria living in extreme conditions as an analog to ancient Earth or modern extraterrestrial conditions [145]. What has driven many of these studies is the incorporation of new techniques or methodologies that either allow deeper elucidation of known systems, or the discovery of novel systems. While these techniques and methodologies were originally intended for use in biological systems, there are clear applications towards prebiotic systems as well. Here, we will describe some of the state-of-the-art technological developments recently made by biophysicists in Japan, and how these techniques could contribute towards OoL research.

Perhaps, there is no one technology that has contributed more to knowledge on evolutionary dynamics than next generation sequencing [146]. From the humble beginnings of Sanger Sequencing [147], current sequencing techniques can now sequence the full human genome on the order of hours or days. Leveraging the high-throughput capability of sequencing allowed the explosion of *in vitro* RNA evolution experiments [113], resulting in massive amounts of information

on RNA evolution dynamics within a very short period of time [148,149]. Accurate sequence quantification methodologies developed by Katsuyuki Shiroguchi's lab at the RIKEN Quantitative Biology Center [150,151], and single molecule sequencing techniques developed by Masateru Taniguchi's lab at Osaka University [152], now allow analysis of RNAs, or even peptides [153], at single molecule resolution, which could be extended for *in situ* high-resolution analyses of primitive evolving genetic or catalytic systems for life detection missions, or even non-biological scaffolding systems present in the prebiotic milieu, such as polyesters [15].

Although sequences do not lie, the adage "seeing is believing" still suggests that images are still powerful when describing biological phenomena. To that end, state-of-the-art imaging systems have pioneered new developments and knowledge in these fields. As mentioned above, researchers at Kanazawa University have developed high speed AFM machinery capable of sub-second resolution [154,155]. This time-resolution could allow further elucidation into the assembly dynamics and topology of primitive self-assembling systems, such as fiber-forming tripeptides [20], lipid layers on mineral surfaces [156], or complex-folded RNA origamis [157]. Advances in cryo-EM [158] or other novel crystallography methods [159] pioneered by many research labs in Japan could allow for native structural elucidation of reaction dynamics within protocells [160,161], ribonucleopeptides [162,163], or primitive compartments [164]. Finally, developments in stimulated Raman scattering microscopy [165,166] and superresolution microscopy [167,168] applied towards protocellular studies could further provide spatial insights of reactions within primitive protocells with high resolution.

Coupling these imaging and sequencing advances with novel microfluidics and nanotechnology would be the next important avenue to consider for OoL researchers. Further development of nanopore technology (used in the single molecule sequencing methods described above [152]) [169,170] would further allow analysis of a wider range of prebiological polymer systems, while additionally opening up the possibility of the incorporation of sequencing technologies onto *in situ* life detection missions [171]. While microfluidics technologies have already been incorporated into sequencing technologies [172], recent advances in coupling microfluidic systems with compartment formation and manipulation have been developed by Masahiro Takinoue's lab at Tokyo Institute of Technology [173] and Hiroyuki Noji's lab at University of Tokyo [174,175]. These advances allow the high-throughput generation of membraneless droplet compartments or liposomes in a size-controlled manner, allowing not only further downstream screening for biomedical or biotechnological purposes, but also potential incorporation into *in vitro* evolutionary systems, such as liposome display [129]. High-throughput generation of model protocells would allow researchers to more easily and more efficiently study the assembly and dynamics of primitive compartment systems, while combining new high-throughput microfluidics technologies with sequencing techniques could even further push our understanding of primitive evolving systems [176,177].

These pioneering and emerging technologies represent only a flavor or subset of the wide cache of significant technologies developed by and available to biophysicists in Japan and around the world. The biophysics research field in Japan is at the forefront of pioneering state-of-art instrumentation and methodologies to study biological systems, as evidenced by the litany of novel technologies mentioned herein (and many not mentioned), which is also supported by many biological analytical instrumentation companies in Japan who have an interest in moving beyond the state-of-the-art in analytical instrumentation. Thus, there is a clear synergy to harness this wealth of technological and methodological development towards prebiotic systems as well. There is also a current trend in our research field in Japan to internationalize research in order to promote more fruitful collaborations where each side can contribute their own strengths. These include running the Annual Meeting of the Biophysical Society in Japan entirely in English (one of the first scientific societies in Japan to do so) [111], offering both English and Japanese biophysics journals run by the Biophysical Society of Japan, the creation of international research laboratories, such as through the World Premier International Research Center Initiative (WPI) [178] or the Laboratory for Integrated Micro-Mechanical

Systems (LiMMS) at University of Tokyo (which is actually sponsored in part by the French CNRS (Centre National de la Recherche Scientifique)) [179], and also the existence of many research grants promoting the international exchange of ideas, resources, and students (or postdocs) through the Japan Society for the Promotion of Science (JSPS) [180,181] or the Japan Science and Technology Agency (JST) [182]. We believe that the future of OoL research in Japan, as well as around the world, will stem from major technological and methodological advances and scientific discoveries coupled with international and interdisciplinary collaborations led by biophysicists, providing a clear synergy to current (Hayabusa2 [183]) and upcoming (MMX [49]) *in situ* sample return life detection missions being led by JAXA or other international space missions, such as OSIRIS-REx led by NASA [184]. It will be up to the biophysicists to lend these technologies and target the important unanswered problems in OoL. With the continued collaboration of biophysicists together with biologists, chemists, geologists, and planetary scientists, the future of OoL research looks bright indeed.

**Author Contributions:** T.Z.J. and Y.K. wrote the manuscript.

**Funding:** T.Z.J. is a research fellow from ELSI at Tokyo Institute of Technology, and is supported by a Tokyo Institute of Technology Seed Grant ("Tane" 1798), a Kakenhi Grant-in-Aid for Scientific Research from the Japan Society for the Promotion of Science (JSPS) Grant-in-Aid for Early-Career Scientists (JP18K14354), and a project grant from the Japan Astrobiology Center (ABC) (AB311021). Y.K. is supported by JSPS grants 16H06156 and 26106003 as well as grant JPMJPR18K5 from the Japan Science and Technology Agency. ELSI is a member of the World Premier International Research Center Initiative and is supported in part by the Japanese Ministry of Education, Culture, Sports, Science and Technology.

**Acknowledgments:** The authors would like to acknowledge the staff and researchers from the Earth-Life Science Institute (ELSI) for helpful discussions. The authors plan to host a symposium soon, related to how biophysicists are poised to be the future leaders of the OoL field.

**Conflicts of Interest:** No conflicts of financial interests are declared.

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
