# Peer review of "Recent Advances in Origins of Life Research by Biophysicists in Japan"

_challenges, doi:10.3390/challe10010028_

Reviewer 1 Report

This is a very well written and comprehensive review. All statements are supported with proper references. I suggest the following reference as a good fit to be added to this review: "A synthetic molecular system capable of mirror-image genetic replication and transcription." 

It reports the synthesis of the mirror image of an important biological molecule.

Overall, I find this paper publishable in present form. 

Author Response

This is a very well written and comprehensive review. All statements are supported with proper references. I suggest the following reference as a good fit to be added to this review: "A synthetic molecular system capable of mirror-image genetic replication and transcription." It reports the synthesis of the mirror image of an important biological molecule. Overall, I find this paper publishable in present form.

We thank the reviewer for these comments and for their support of our paper, and have added this reference in addition to a brief comment about the origin of homochiral systems highlighted in yellow.

Reviewer 2 Report

In the review “Recent Advances in Origin of Life Research by Biophysicists in Japan” by Jia et al. address and highlight quite important contributions by Japanese researchers to the origins of life field (OoL).

Authors have touched different aspects of the interdisciplinary OoL problem from the synthesis of biopolymers, compartmentalization, and biopolymer evolution in a very thorough manner. However, they have not taken into account the contributions from theoretical studies addressing the OoL problem. There is a huge body of work on, but not limited to, ‘emergence of early replicators’, ‘autocatalytic sets’ and their ‘evolution’. Authors can look up the work from Doron Lancet, Eörs Szathmáry, Sandeep Krishna, Wim Hordijk, and Mike Steel. Also, Paul Higgs, Jeremy England, and others have contributed to addressing different aspects of OoL problem. It will be quite good to highlight the contributions of Japanese researchers in the developing theoretical framework for OoL problem, e.g. Kunihiko Kaneko. A separate section, even a small one, on such studies will make the review complete highlight all aspects of OoL research.

Rest, authors have pretty well covered all the important work in the field and cited the relevant papers. I highly recommend the review for the publication.

Here are some minor comments:

line 166: remove “now”

line 173: what do authors mean by “first biochemistries”. Biochemistry term is as such used for a branch of science. It would be odd to think ‘first biochemistries’. I think would be appropriate to say ‘first biochemical regulatory interactions’, if it is what authors intended to say.

line 194: clarify “allowing for selective genetic evolution within the encapsulated space”. Is there any example of ‘selective’ genetic evolution in a compartment? Why would be evolution within a compartment advantageous? Information in the form of a polymer or network in the compartment and selective advantage of one compartment over others would be imaginable but within a compartment is not clear. 

Line 290: give an example of ‘membraneless polyester microdroplets’

Line 305: correct ‘selective evolution’ to ‘selection’

Line 332-334: In the work by Mizuuchi et al. the authors clearly differentiated the word ‘reproduction’ from ‘replication’ (Mizuuchi 2019 Life, p2, para 1). So authors here should change the replicatory networks to “RNA networks capable of reproduction (self- or cross-) and forming collectively autocatalytic sets”.

Line 441: Microfluidics combined with sequencing technologies. Authors should also refer to ‘DropSeq (https://doi.org/10.1016/j.cell.2015.05.002)’ and ‘inDrop (https://www.cell.com/cell/fulltext/S0092-8674(15)00500-0)’ here.

Author Response

Reviewer 2

In the review “Recent Advances in Origin of Life Research by Biophysicists in Japan” by Jia et al. address and highlight quite important contributions by Japanese researchers to the origins of life field (OoL).

We thank the reviewer for reading this manuscript and for understanding the importance of contributions by researchers in Japan to the Origins of Life field.

Authors have touched different aspects of the interdisciplinary OoL problem from the synthesis of biopolymers, compartmentalization, and biopolymer evolution in a very thorough manner. However, they have not taken into account the contributions from theoretical studies addressing the OoL problem. There is a huge body of work on, but not limited to, ‘emergence of early replicators’, ‘autocatalytic sets’ and their ‘evolution’. Authors can look up the work from Doron Lancet, Eörs Szathmáry, Sandeep Krishna, Wim Hordijk, and Mike Steel. Also, Paul Higgs, Jeremy England, and others have contributed to addressing different aspects of OoL problem. It will be quite good to highlight the contributions of Japanese researchers in the developing theoretical framework for OoL problem, e.g. Kunihiko Kaneko. A separate section, even a small one, on such studies will make the review complete highlight all aspects of OoL research.

We thank the reviewer for this suggestion and add a separate section focusing on the contribution of theoretical researchers towards OoL including Kunihiko Kaneko and Takashi Ikegami.

Rest, authors have pretty well covered all the important work in the field and cited the relevant papers. I highly recommend the review for the publication.

We thank the reviewer for recognizing the worth of our manuscript and recommending publicaiton.

Here are some minor comments:

line 166: remove “now”

We have removed this word.

line 173: what do authors mean by “first biochemistries”. Biochemistry term is as such used for a branch of science. It would be odd to think ‘first biochemistries’. I think would be appropriate to say ‘first biochemical regulatory interactions’, if it is what authors intended to say.

We have changed this wording.

line 194: clarify “allowing for selective genetic evolution within the encapsulated space”. Is there any example of ‘selective’ genetic evolution in a compartment? Why would be evolution within a compartment advantageous? Information in the form of a polymer or network in the compartment and selective advantage of one compartment over others would be imaginable but within a compartment is not clear.

Perhaps we misworded this statement, and in fact we meant to write “selective evolution of genetic replicators”. This is because when confined within the same compartment, advantageous replicators would replicate only themselves (and descendant, but similar, replicators), resulting in further proliferation (and evolution) of themselves into the dominant entity. However, in solution highly efficient replicators could potentially catalyze the replication of unrelated molecules with no replication activity, conferring no advantage to themselves, resulting in stunted replication, and hence evolution. Szostak, Bartel, and Luisi’s comment (also cited in our manuscript in the next line) provide a good description of this process, and thus we have changed the wording to be slightly more nuanced, while moving the reference to directly after this statement.

The point we were trying to illustrate was the fact that a compartment would allow an advantageous genetic (or catalytic) polymer to remain confined to the same local space and continue to proliferate and evolve, whereas

Line 290: give an example of ‘membraneless polyester microdroplets’

We have now added a reference to the abstract book of the 2018 Biophysical Society of Japan Annual Meeting, where polyester microdroplets were first reported.

Line 305: correct ‘selective evolution’ to ‘selection’

We have changed this wording.

Line 332-334: In the work by Mizuuchi et al. the authors clearly differentiated the word ‘reproduction’ from ‘replication’ (Mizuuchi 2019 Life, p2, para 1). So authors here should change the replicatory networks to “RNA networks capable of reproduction (self- or cross-) and forming collectively autocatalytic sets”.

We have changed this wording.

Line 441: Microfluidics combined with sequencing technologies. Authors should also refer to ‘DropSeq (https://doi.org/10.1016/j.cell.2015.05.002)’ and ‘inDrop (https://www.cell.com/cell/fulltext/S0092-8674(15)00500-0)’ here.

We have added these references, as long as a unifying statement that discusses the combination of microfluidics techniques with sequencing technologies.

Reviewer 3 Report

Review of challenges-481395

Recent Advances in Origin of Life Research by Biophysicists in Japan

by Tony Z. Jia and Yutetsu Kuruma

The manuscript explores how Japanese biophysicists (or biophysical research in Japan) contributed to the understanding of the origins and early evolution of life. The manuscript covers a wide range of biophysical topics (not only the authors’ specific research fields), and does a good job to announce how powerful Japanese biophysical teams are. I personally enjoyed reading and learnt a lot from the manuscript. There are several concerns to address before publication, however, I recommend the manuscript for publication in the journal Challenges once the authors carefully address the followings.

Major points

1.      Figure 1 is slightly misleading because it seems to suggest each problem listed in the figure can be clearly categorized in different fields. However, this is not accurate. (As the authors probably know,) for example, “assembly of membrane compartments” in “Novel Biophysics Techniques” should also to some extend depend on inherent “chemistry” of particular types of lipids; some studies suggested that “geological” environments (e.g., mineral surfaces) promoted membrane formation. “First cells” might have fully relied on “genetics”, but they might have cell-divided fully “physically”. So, I suggest A) the authors clearly mention the category is just a “main field”, or B) just mention what kind of problems “Novel Biophysics Techniques” can address.

2.      In the legend of Figure 1, the authors stated “Specifically, biophysicists in Japan have been contributing towards… ; these are believed to be the most important steps in the origin of life.” But I think what the most important steps in the origins of life depend on researchers. I suggest the authors cite an appropriate paper if you claim the sentence, or just say “…believed to be important steps…”.

3.      I do not understand why the authors highlighted the work of Figure 4. It is more straightforward to highlight a work done by Japanese researcher(s) because the aim of the paper is to review biophysical research in Japan.

4.      L350 “with the identical sequence as the initial host RNA” is misleading, because 1) they are not identical (as authors mentioned, the parasitic RNAs lack the replicase coding region); and 2) parasitic sequences vary (the Ichihashi’s work analyzed just one type of (=most dominant) sequence). I therefore suggest that the authors rewrite the sentence. For example, with “an” identical sequence “region” as the host RNA.

5.      L363 “while even transient rather than sustained compartmentalization could have even assisted in parasite resistance [66]” is not accurate in this context. A transient compartmentalization can assist parasite resistance only for self-replicating (single) RNA molecule, but not a replicating network. The work [121] showed that a transient well-mixed stage (inevitable in a transient compartmentalization as introduced in [66]) would quickly disrupt the replication network (because each component is difficult to be re-encapsulated in the same compartment again); instead, a heterogeneous mixing (or “sustained compartmentalization”) can make the network sustainable. I therefore suggest the authors rephrase or delete the part “while even…resistance [66]”, or move it to a different place.

6.      The authors mainly used “origins of life” throughout the paper, but they also used “origin of life” at L2, 95, and 99. The authors should be consistent in the usage of this word choice, which should represent the authors’ philosophy. This is particularly important in the field of the origins of life because it reflects how the authors recognize the emergence of life: the single origin or multiple origins, etc.

7.      How does the authors rank Japanese biophysical research? I am not sure if this need to be incorporated into the current manuscript (thus it is up to authors), but it may be worth describing more about the current scientific positions, uniqueness, and novelty of biophysical research in Japan “on a global level”. Are there specific advantages (or disadvantages if any) in biophysical research in Japan compared to other countries? I think such information would contribute to collaborating with other countries and recruiting international students or postdocs to the biophysical research community in Japan, and therefore fostering science in Japane.

Minor comments

L23: “to not only use…but also to consider” should be “not only to use…”

L53: Something is wrong with “proceed”, which is an intransitive verb. Maybe misspelling of “precede”, but it basically means before (should not be right as well).

L54: should add “could have” before “replicated non-enzymatically”

L78: The usage of “However” in this position does not make sense. I would simply remove it.

L79: using “still” is strange; it might also be better to use “could have been” instead of “could (still) be”

L80: using “still” is strange.

L83: the sentence “Pre “life” can still use biophysical tools” does not make sense because “we” use biophysical tools.

L91: “prospective” is probably a misspelling of “perspective”.

L95: “O” of “Origin” should be a lower character

L125: “provide” should be “provided”

L138: the authors should add a reference after “stabilizing”. The authors might have meant this is the study [47] by Yoshihiro Furukawa, but to me, it is unclear (in that case).

L144: what is “ribose minerals”? Mistake?

L175: require “studies on” after “but” in the current grammatical context

L181: add “Some” before “RNA”

L194: add some reference(s) after “encapsulated space”

L239: “which leads to” should be “which could lead for”

L267: “lab” should be “laboratory”

L290: delete a space immediately before “,”

L292: “to not only” should be “not only for”

L309: The citation of [113] in “is beyond the scope of this review [113]” is ambiguous. If it is a general review of in vitro selection, you should write so (or the authors can just remove the citation because the authors said it is beyond the scope.)

L344: “are” should be “is” (or “has been”).

L376: the sentence “Many in vitro…” lacks a conjunction that connects the two sentences (“Many in vitro…evolution” and “these are…proteins”).

L396: delete “us”

L397: “which” should be “that” in this context

L402: delete a space after [38,39]

Author Response

Reviewer 3

The manuscript explores how Japanese biophysicists (or biophysical research in Japan) contributed to the understanding of the origins and early evolution of life. The manuscript covers a wide range of biophysical topics (not only the authors’ specific research fields), and does a good job to announce how powerful Japanese biophysical teams are. I personally enjoyed reading and learnt a lot from the manuscript. There are several concerns to address before publication, however, I recommend the manuscript for publication in the journal Challenges once the authors carefully address the followings.

We thank the reviewer for reading our manuscript and appreciate the very detailed comments that were provided. We believe that the manuscript as a whole has certainly improved with these comments.

Major points

1.      Figure 1 is slightly misleading because it seems to suggest each problem listed in the figure can be clearly categorized in different fields. However, this is not accurate. (As the authors probably know,) for example, “assembly of membrane compartments” in “Novel Biophysics Techniques” should also to some extend depend on inherent “chemistry” of particular types of lipids; some studies suggested that “geological” environments (e.g., mineral surfaces) promoted membrane formation. “First cells” might have fully relied on “genetics”, but they might have cell-divided fully “physically”. So, I suggest A) the authors clearly mention the category is just a “main field”, or B) just mention what kind of problems “Novel Biophysics Techniques” can address.

We thank the reviewer for this comment. It is very true that that these problems should be categorized in one specific field. Rather, we have highlighted either the “main field” or the field where traditionally the most advances have been made. We have modified the text and the figure caption to reflect this, and also add a statement that explains that these are not hard boundaries.

2.      In the legend of Figure 1, the authors stated “Specifically, biophysicists in Japan have been contributing towards… ; these are believed to be the most important steps in the origin of life.” But I think what the most important steps in the origins of life depend on researchers. I suggest the authors cite an appropriate paper if you claim the sentence, or just say “…believed to be important steps…”.

Coming from specific sub-fields within OoL research leads to researchers sometimes having tunnel-vision about specific topics and which ones are most important. We really appreciate the reviewer for pointing out this fact, especially as we perhaps have not fully appreciated the contributions or importance of researchers from other fields, and thus we have changed the wording.

3.      I do not understand why the authors highlighted the work of Figure 4. It is more straightforward to highlight a work done by Japanese researcher(s) because the aim of the paper is to review biophysical research in Japan.

We thank the reviewer for this comment, and now have replaced the figure with one that includes an original author-drawn schematic, as well as a microscope image from a publication by Tokyo Tech researcher Tony Z. Jia.

4.      L350 “with the identical sequence as the initial host RNA” is misleading, because 1) they are not identical (as authors mentioned, the parasitic RNAs lack the replicase coding region); and 2) parasitic sequences vary (the Ichihashi’s work analyzed just one type of (=most dominant) sequence). I therefore suggest that the authors rewrite the sentence. For example, with “an” identical sequence “region” as the host RNA.

We thank the reviewer for this comment, and now have modified the text to be more correct.

5.      L363 “while even transient rather than sustained compartmentalization could have even assisted in parasite resistance [66]” is not accurate in this context. A transient compartmentalization can assist parasite resistance only for self-replicating (single) RNA molecule, but not a replicating network. The work [121] showed that a transient well-mixed stage (inevitable in a transient compartmentalization as introduced in [66]) would quickly disrupt the replication network (because each component is difficult to be re-encapsulated in the same compartment again); instead, a heterogeneous mixing (or “sustained compartmentalization”) can make the network sustainable. I therefore suggest the authors rephrase or delete the part “while even…resistance [66]”, or move it to a different place.

We thank the reviewer for this comment, and now have deleted this.

6.      The authors mainly used “origins of life” throughout the paper, but they also used “origin of life” at L2, 95, and 99. The authors should be consistent in the usage of this word choice, which should represent the authors’ philosophy. This is particularly important in the field of the origins of life because it reflects how the authors recognize the emergence of life: the single origin or multiple origins, etc.

We thank the reviewer for this comment, and now have modified the sections specified to be consistent in using “origins” instead of “origin”. In one case, we modified the word “origin” to “emergence” to decrease ambiguity, when referring to modern biomolecules, since there is still only one (known) set of modern biomolecules.

7.      How does the authors rank Japanese biophysical research? I am not sure if this need to be incorporated into the current manuscript (thus it is up to authors), but it may be worth describing more about the current scientific positions, uniqueness, and novelty of biophysical research in Japan “on a global level”. Are there specific advantages (or disadvantages if any) in biophysical research in Japan compared to other countries? I think such information would contribute to collaborating with other countries and recruiting international students or postdocs to the biophysical research community in Japan, and therefore fostering science in Japane.

We thank the reviewer for this very insightful comment. Truly, Challenges is a very unique journal in that its scope allows the writer to discuss or propose solutions to various problems or issues in, for example, research administration, research collaboration, etc. As such, we have taken this opportunity to briefly add some more into the final paragraph to highlight some of the ways by which biophysics in Japan is novel, and also show some of the internationalization efforts of this field. Again, we thank the reviewer for this insightful comment and for the opportunity to include these ideas in this manuscript.

Minor comments

L23: “to not only use…but also to consider” should be “not only to use…”

We have modified the text for the parallelism to be grammatically correct.

L53: Something is wrong with “proceed”, which is an intransitive verb. Maybe misspelling of “precede”, but it basically means before (should not be right as well).

We have changed the wording to “followed” in order to be less ambiguous.

L54: should add “could have” before “replicated non-enzymatically”

We have made this correction.

L78: The usage of “However” in this position does not make sense. I would simply remove it.

We have made this correction.

L79: using “still” is strange; it might also be better to use “could have been” instead of “could (still) be”

L80: using “still” is strange.

In both cases, we have removed the word “still”.

L83: the sentence “Pre “life” can still use biophysical tools” does not make sense because “we” use biophysical tools.

We meant that “Researchers studying” pre-life…, and  have made this correction.

L91: “prospective” is probably a misspelling of “perspective”.

“Prospective” is indeed correct. While “perspective” means an opinion, “prospective” means the forecast looking forward, which is the intended uses in this situation.

L95: “O” of “Origin” should be a lower character

We have made this correction. 

L125: “provide” should be “provided”

We have made this correction.

L138: the authors should add a reference after “stabilizing”. The authors might have meant this is the study [47] by Yoshihiro Furukawa, but to me, it is unclear (in that case).

We have moved the reference to the end of the sentence, as it is relevant to the entire statement.

L144: what is “ribose minerals”? Mistake?

We meant to write “ribose-binding minerals” and have made this correction.

L175: require “studies on” after “but” in the current grammatical context

We have instead now modified this sentence to read “but also other prebiotically…” to maintain correct grammatical construction.

L181: add “Some” before “RNA”

We have made this correction.

L194: add some reference(s) after “encapsulated space”

We have slightly modified this sentence at the suggestion of another reviewer (although the main idea remains the same), and have added a reference by Szostak, Bartel, and Luisi which comments on this statement.

L239: “which leads to” should be “which could lead for”

We have rewritten this as “which could lead towards”.

L267: “lab” should be “laboratory”

We have made this correction.

L290: delete a space immediately before “,”

We have made this correction.

L292: “to not only” should be “not only for”

We now change the dependent clause to maintain the original parallel structure by moving “to” before “also”. Perhaps the reviewer mistook the word “further” for a noun, when in fact it is being used as a verb, and as such “to” is a correct preposition.

L309: The citation of [113] in “is beyond the scope of this review [113]” is ambiguous. If it is a general review of in vitro selection, you should write so (or the authors can just remove the citation because the authors said it is beyond the scope.)

We have made this correction and deleted this reference.

L344: “are” should be “is” (or “has been”).

We have made this correction.

L376: the sentence “Many in vitro…” lacks a conjunction that connects the two sentences (“Many in vitro…evolution” and “these are…proteins”).

We have made this correction and added in “Although” before the start of the main clause.

L396: delete “us”

We have made this correction.

L397: “which” should be “that” in this context

We have made this correction.

L402: delete a space after [38,39]

We have made this correction.